**Data Availability Statement:** All data are freely available within the manuscript itself.

# FAST and Agile–the MASLD drift: Validation of Agile 3+, Agile 4 and FAST scores in 246 biopsy-proven NAFLD patients meeting MASLD criteria of prevalent caucasian origin

**Madalina-Gabriela Taru**[1,2,3‡], **Cristian Tefas**[1,2‡], **Lidia Neamti**[1,2], **Iulia Minciuna**[1,2], **Vlad Taru**[1,2,4], **Anca Maniu**[2], **Ioana Rusu**[2], **Bobe Petrushev**[2], **Lucia Maria Procopciuc**[1], **Dan Corneliu Leucuta**[1], **Bogdan Procopet**[1,2], **Silvia Ferri**[5], **Monica Lupsor-Platon**[1,2]*, **Horia Stefanescu**[2]

1 Faculty of Medicine, "Iuliu Hatieganu" University of Medicine and Pharmacy, Cluj-Napoca, Cluj, Romania, 2 Regional Institute of Gastroenterology and Hepatology "Octavian Fodor", Cluj-Napoca, Cluj, Romania, 3 Department of Medical and Surgical Sciences, University of Bologna, Bologna, Italy, 4 Division of Gastroenterology and Hepatology, Department of Medicine III, Medical University of Vienna, Vienna, Austria, 5 Division of Internal Medicine, Hepatobiliary and Immunoallergic Diseases, IRCCS Azienda Ospedaliero—Universitaria di Bologna, Bologna, Italy

‡ MGT and CT are equally to this work as shared first authorship.
* monica.lupsor@umfcluj.ro

## Abstract

### Background

MASLD is a prevalent chronic liver condition with substantial clinical implications. This study aimed to assess the effectiveness of three new, elastography-based, scoring systems for advanced fibrosis ≥F3 (Agile 3+), cirrhosis F4 (Agile 4), and fibrotic NASH: NASH + NAS ≥4 + F≥2 (FAST score), in a cohort of biopsy-proven NAFLD meeting MASLD criteria. Our secondary aim was to compare their diagnostic performances with those of other fibrosis prediction tools: LSM-VCTE alone, and common, easily available scores (FIB-4 or APRI).

### Methods

Single-center, retrospective study, on consecutive patients with baseline laboratory tests, liver biopsy, and reliable LSM-VCTE measurements. The discrimination between tests was evaluated by analyzing the AUROCs. Dual cut-off approaches were applied to rule-out and rule-in ≥F3, F4 and fibrotic NASH. We tested previously reported cut-off values and provided our best thresholds to achieve Se ≥85%, Se ≥90%, and Sp ≥90%, Sp ≥95%.

### Results

Among 246 patients, 113 (45.9%) were women, and 75 (30.5%) presented diabetes. Agile 3 + and Agile 4 demonstrated excellent performance in identifying ≥F3 and F4, achieving AUROCs of 0.909 and 0.968, while the FAST score yielded acceptable results in distinguishing fibrotic NASH. When compared to FIB-4 and LSM-VCTE, both Agile 3+ and Agile 4

**Funding:** This work was partially funded from the grants number PN-III-P4-PCE-2021-1140 from UEFISCDI, the Romanian Executive Agency for Higher Education, Research, Development, and Innovation Funding and 35167/17.12.2021 from the University of Medicine and Pharmacy "Iuliu Haţieganu" Cluj-Napoca. Madalina-Gabriela Taru is financed by The Study Loans and Scholarships Agency, The Ministry of Education, Romania.

**Competing interests:** NO authors have competing interests.

**Abbreviations:** Acc, Accuracy; ACLD, advanced chronic liver disease; Alb, albumin; ALP, alkaline phosphatase; ALT, alanine-aminotransferase; APRI, AST to Platelet Ratio Index; AST, aspartate-aminotransferase; AUROC, area under the receiver operating curve; BMI, body mass index; CAP, controlled attenuation parameter; CI, confidence interval; CLD, chronic liver disease; CRN, clinical research network; DM, diabetes mellitus; ELF, Enhanced Liver Fibrosis score; F, fibrosis; F0, absence of fibrosis; F1, mild fibrosis; F2, significant fibrosis; F3, advanced fibrosis; F4, cirrhosis; FLIP, fatty liver inhibition of progression; FAST, FibroScan-AST score; FIB-4, Fibrosis 4 index; GGT, gamma glutamyl transpeptidase; Hb, hemoglobin; HCC, hepatocellular carcinoma; Hz, hertz; INR, international normalized ratio; IQR, interquartile range; kPa, kilopascals; LB, liver biopsy; LDL, low-density lipoprotein cholesterol; LREs, liver-related events; LS, liver stiffness; LSM, liver stiffness measurement; M, mean; MASH, metabolic disfunction-associated steatohepatitis; MASLD, metabolic disfunction-associated steatotic liver disease; n, number; NAFLD, nonalcoholic fatty liver disease; NAS, NAFLD activity score; NASH, nonalcoholic steatohepatitis; NITs, noninvasive tests; NPV, negative predictive value; PPV, positive predictive value; Q1, percentile 25; Q3, percentile 75; SD, standard deviation; Se, sensitivity; Sp, specificity; T2DM, type 2 diabetes mellitus; TB, total bilirubin; Tg, triglycerides; Tot Cho, total cholesterol; VCTE, vibration controlled transient elastography.

performed better than FIB-4 and had a similar performance to LSM-VCTE, but with higher diagnostic accuracy, hence reducing the grey zone.

## Conclusion

Agile 3+ and Agile 4 are reliable, non-invasive tests for identifying advanced fibrosis or cirrhosis in MASLD patients, while FAST score demonstrates moderate performance in identifying fibrotic NASH.

## Introduction

Non-alcoholic fatty liver disease (NAFLD), also referred to by the newly defined term metabolic dysfunction-associated steatotic liver disease (MASLD) [1] is the most prevalent chronic liver condition worldwide (estimated to affect up to 38% of the entire population) [2]. Liver fibrosis is a crucial determinant of prognosis in patients with MASLD [3], leading to a significant rise in overall mortality and increased risk of developing liver-related events (LREs), especially among patients with advanced fibrosis (≥F3) or cirrhosis (F4) [4].

Liver biopsy (LB) is currently the accepted standard for evaluating liver fibrosis. However, it is hampered by its invasive nature, intra and inter-observer variability and sampling errors [5–7]. Given these limitations, the most straightforward approach to identify MASLD patients with suspected advanced chronic liver disease (ACLD) would involve applying non-invasive tests (NITs), while concurrently striving for cost-efficiency [8]. The utilization of NITs, which can be easily repeated over time and offer the potential to compare successive measurements, could improve the overall care of patients with chronic liver disease (CLD) in general and of those with MASLD, in particular [9].

Agile 3+ and Agile 4 are two scores comprising clinical and laboratory factors (including AST/ALT ratio, platelet count, gender, diabetes status, and age for Agile 3+) along with liver stiffness measurement (LSM) using vibration-controlled transient elastography (VCTE) [10]. These scores have been developed to predict advanced fibrosis (≥F3) and cirrhosis (F4), respectively, in patients with NAFLD [10]. As established non-invasive tests (NITs), LSM-VCTE, and FIB-4 were demonstrated to have good performance in ruling-out advanced fibrosis in people with NAFLD [11]. The newly developed Agile 3+ and Agile 4 scores were meant to provide higher positive predictive values (PPVs) for ruling-in ≥F3 and F4, and to reduce the number of indeterminate results [10]. These scores correlate well with the severity of liver fibrosis, decrease the number of patients left in the so-called "grey zone", and increase the PPV for ruling-in ≥F3 and F4, respectively [10].

The FibroScan-AST (FAST) score has been developed in 2020 for the non-invasive identification of patients with non-alcoholic steatohepatitis (NASH), concomitant significant activity (NAS ≥4) and significant fibrosis (F≥2) as per liver biopsy [12].

In this study, we aimed to investigate the effectiveness of Agile 3+, Agile 4 and FAST scores in discriminating advanced fibrosis, cirrhosis, and fibrotic NASH, respectively, in our cohort of biopsy-proven NAFLD patients that met MASLD criteria from a tertiary medical center in Cluj-Napoca, Romania. A secondary goal was to determine if these scores outperformed commonly used NITs such as LSM-VCTE, FIB-4 and APRI (compared to FAST-score) for predicting ≥F3, F4 and fibrotic NASH, respectively.

## Materials and methods

### Patients

This retrospective analysis included 246 consecutive adult patients (18–80 years old), evaluated for suspected NAFLD, from our tertiary care center in Cluj-Napoca, Romania. The recruitment period started on the 1[st] of January 2007 and ended on the 18[th] of July 2023. All included patients had undergone liver biopsy (percutaneous or transjugular) for diagnostic purposes and presented baseline reliable VCTE measurements within a maximum three weeks prior to the liver biopsy. We excluded patients with missing data necessary for calculating the Agile 3+, Agile 4 and FAST scores, missing fibrosis stage on liver biopsy, history of chronic liver disease other than NAFLD (such as viral, cholestatic, immune etc.), high alcohol consumption (defined by >21 drinks, on average, per week in men and >14 drinks, on average, per week in women [13]), and ALT and AST >5 times the upper normal limit.

All patients had the following parameters collected at baseline: age, gender, body mass index (BMI), fasting glucose and history of diabetes, complete blood count, coagulation parameters, liver function profile, renal function, lipidic profile, and serum electrolytes.

This retrospective study of consecutively enrolled patients was conducted in accordance with the principles of the Helsinki Declaration and with the local and national laws. The study protocol was approved by the local institutional review boards—The Ethics Committee from "Iuliu Hatieganu" University of Medicine and Pharmacy", Cluj-Napoca, Romania, (PN-III-P4-PCE-2021-1474 study—number of approval AVZ259/14.09.2022). The informed consent was signed by all participants at the moment of the enrollment. Data was accessed for research purposes on the 20[th] of September 2023. The authors did not have access to information that could identify individual participants during and after data collection.

### Liver biopsy

Liver biopsies were fixed in formalin and embedded in paraffin. Histopathological staging for liver fibrosis was performed according to the NASH Clinical Research Network (CRN) scoring system and served as the reference standard [5]. Steatosis (0–3), ballooning (0–2) and inflammation (0–3) were also scored using the NASH CRN scoring system [5]. One pathologist specialized in liver diseases, blinded to the NITs results, staged fibrosis on the biopsy specimens, as: stage 0—absence of fibrosis (F0), stage 1—perisinusoidal or portal (F1), stage 2—perisinusoidal and portal/periportal (F2), stage 3—septal or bridging fibrosis (F3), stage 4—cirrhosis (F4). The NAFLD activity score (NAS) was calculated as the sum of steatosis, ballooning, and lobular inflammation grades and ranged from 0 to 8 [5]. NASH was defined on LB as the presence of steatosis, hepatocyte ballooning, and lobular inflammation with at least 1 point for each category (FLIP-NASH) [14], following the seminal study on FAST score [12]. Every biopsy specimen included in the analysis was taken from the right lobe (percutaneous or transjugular) and had a minimum of 6 portal tracts.

### Fibrosis prediction formulas

The Fibrosis-4 index (FIB-4) was calculated as follows: $FIB4 = \frac{Age(years)x\ AST(U/L)}{PLT(10^9/l)x\ \sqrt{ALT(U/L)}}$ [15];

The AST to platelet ratio index (APRI) was calculated as follows: $APRI = \frac{AST\ level(/ULN)}{PLT(10^9/L)}\ x\ 100$ [16];

### Liver stiffness measurement by vibration controlled transient elastography for staging fibrosis

VCTE (FibroScan, Echosens, Paris, France) was performed by two experienced operators, blinded to the biopsy results, with both M (3.5 Hz frequency), and XL (2.5 Hz frequency)

probes, according to the EASL-ALEH recommendations [11, 17] and considering the integrated automatic probe selection software. Measurements were performed in a fasting state. We considered reliable results as being those representing the mean of 10 valid measurements with an IQR/M below 30%.

## Controlled attenuation parameter by vibration controlled transient elastography for grading steatosis

CAP measurements (available in our clinic since 2012) were performed by FibroScan (Echosens, Paris, France) by two experienced operators, blinded to the biopsy results, simultaneously with LSM and by respecting the principles of CAP measurement [18]. CAP was computed only when the associated LSM was valid and using the same signals as the ones used to measure liver stiffness. Therefore, both stiffness and CAP were obtained during the same examination and in the same volume of liver parenchyma. We considered reliable results those representing the mean of 10 valid measurements with an IQR/M below 30%. The final CAP value was expressed in dB/m.

### Agile 3+, Agile 4 and FAST scores

We calculated the Agile 3+, Agile 4 and FAST scores based on the baseline characteristics for each patient, considering diabetes status: yes = 1, no = 0 and gender: male = 1, female = 0, by using the following formulas [10, 12]:

For Agile 3+:

$$Agile\ 3+ = \frac{e^{logit(p_{F \geq F3})}}{1 + e^{logit(p_{F \geq F3})}}$$

where $logit(p_{F \geq F3}) = -3.92368 + 2.29714 \times ln(LSM) - 0.00902 \times PLT - 0.98633\ x$
$AAR^{-1} + 1.08636 \times Diabetes\ status - 0.38581 \times Sex + 0.03018 \times Age$;

For Agile 4:

$$Agile\ 4 = \frac{e^{logit(p_{F=4})}}{1 + e^{logit(p_{F=4})}},$$

where
$logit(p_{F=4}) = 7.50139 - 15.42498\ x\ \frac{1}{\sqrt{LSM}} - 0.01378 \times PLT - 1.41149 \times AAR^{-1} - 0.53281$;

For FAST score:

$$FAST = \frac{e^{-1.65 + 1.07\ x\ \ln(LSM) + 2.66*10^{-8}\ x\ CAP^3 - 63.3\ x\ AST^{-1}}}{1 + e^{-1.65 + 1.07\ x\ \ln(LSM) + 2.66*10^{-8}\ x\ CAP^3 - 63.3\ x\ AST^{-1}}};$$

### Statistical analysis

Continuous variables were evaluated for their normal distribution with the Kolmogorov-Smirnov test, and then expressed as median with interquartile range (Q1-Q3) or mean with standard deviation (SD), or standard error of mean (SEM). Categorical variables were reported as frequency and percentage. Descriptive statistics were provided for the complete group (n = 246) and for the subgroup of patients for whom FAST score was calculated (n = 136). The one-way ANOVA test was used for intergroup comparison between fibrosis stages for LSM-VCTE, Agile 3+ and Agile 4. The diagnostic performance of Agile 3+, Agile 4 and FAST scores was determined using receiver operating characteristic (AUROC) curves. We calculated the AUROCs with 95% confidence intervals (CI) for the detection of histologically confirmed advanced fibrosis ($\geq$F3), cirrhosis (F4) and fibrotic NASH (NASH + NAS $\geq$4 + F$\geq$2). The

DeLong test was used for comparison of diagnostic performance between Agile scores, LSM only, FIB-4, and APRI. For Agile 3+, Agile 4 and FAST scores, the number of patients remaining in the grey zone was determined. The exact McNemar's test was used to assess the concordance between "grey zones". The statistical significance was considered for p values < 0.05 for all tests. Statistics were performed using the IBM Statistical Package for Social Sciences (SPSS, version 29, IBM Corp., Armonk, NY, USA).

## Results

246 biopsy-proven NAFLD patients were included in the final analysis. The mean number of portal tracts on biopsy was 12 ±8. Of those, 136 presented reliable CAP measurements. Out of 256 patients with reliable VCTE measurements, 4 (1.6%) of them did not meet the MASLD criteria and were not included in the analysis (were considered as lean NAFLD). Fig 1 displays a comprehensive overview of the patient selection process.

The median age at baseline was 52 years (IQR, 20) and median BMI was 29.0 kg/m$^2$ (IQR, 5.1). 113 (45.9%) patients were female and 75 (30.5%) presented diabetes at baseline. Table 1 provides a comprehensive overview of the baseline characteristics for the included patients.

### Diagnostic performance of Agile 3+ and Agile 4 scores

The mean (±SEM) values for LSM-VCTE progressively increased with the increase in fibrosis stages, from F0 (5.6 ± 0.5) kPa, to F1 (6.8 ± 0.3) kPa, F2 (9.0 ±0.5) kPa, F3 (15.5 ± 1.2) kPa and F4 (30.4 ± 3.1) kPa, respectively (Fig 2A).

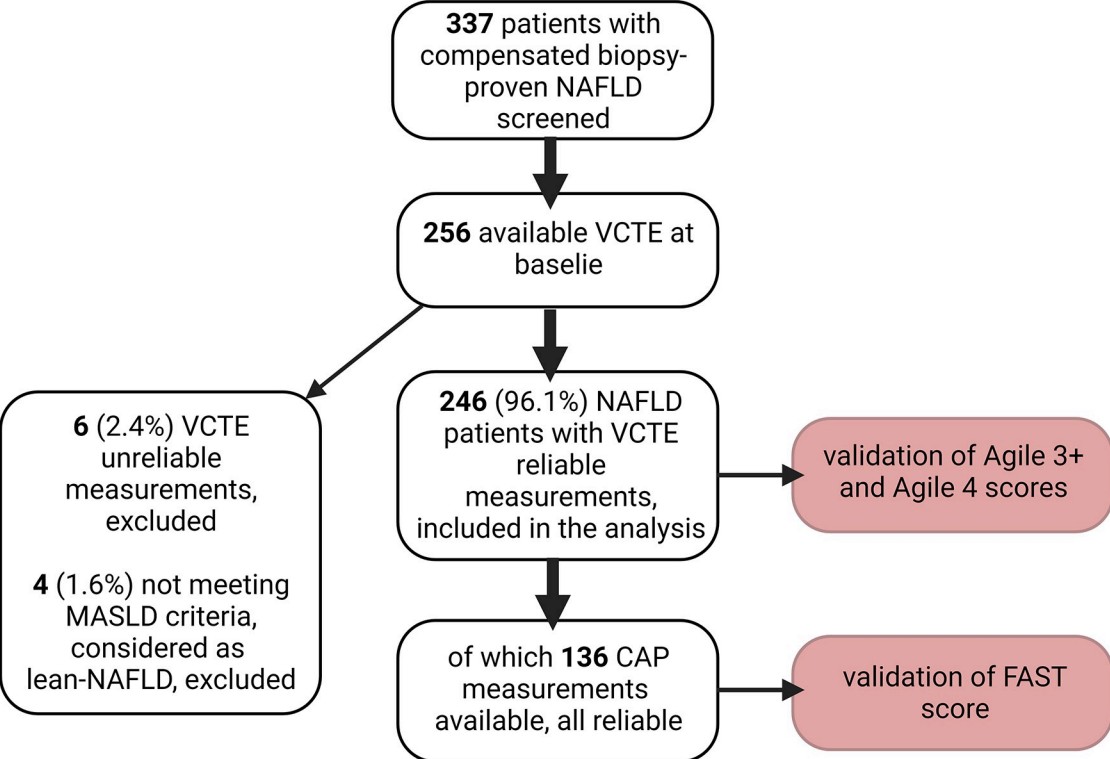

**Fig 1. Flow diagram of patient selection.** NAFLD- nonalcoholic fatty liver disease, VCTE- vibration controlled transient elastography, CAP- controlled attenuation parameter, FAST- FibroScan-AST score.

**Table 1. Baseline characteristics of the patients included in the study.**

| | | Validation of Agile 3+ and Agile 4 scores n = 246 | Validation of FAST score n = 136 |
|---|---|---|---|
| **Variable** | | **Count (%) or Median (Q1-Q3)** | **Count (%) or Median (Q1-Q3)** |
| Age (years) | | 52 (41–61) | 55 (47–63) |
| Gender | Female | 113 (45.9) | 74 (54.4) |
| BMI (kg/m^2) | | 29.0 (27.1–32.2) | 29.0 (27.0–32.5) |
| T2DM | Yes | 75 (30.5) | 49 (36.0) |
| Fibrosis (NASH CRN) | F0 | 25 (10.2) | 4 (2.9) |
| | F1 | 70 (28.4) | 30 (22.1) |
| | F2 | 78 (31.7) | 53 (39.0) |
| | F3 | 44 (17.9) | 29 (21.3) |
| | F4 | 29 (11.8) | 20 (14.7) |
| 0Steatosis (NASH CRN) | S1 | 82 (33.3) | 43 (31.6) |
| | S2 | 85 (34.6) | 50 (36.8) |
| | S3 | 79 (32.1) | 43 (31.6) |
| MASH/NASH | | 222 (90.2) | 114 (83.8) |
| Hb (mg/dl) | | 14.9 (13.6–16.0) | 14.3 (13.1–15.7) |
| Platelets (x10^9/l) | | 230 (187–268) | 224 (174–272) |
| INR | | 1.02 (0.94–1.12) | 1.05 (0.96–1.15) |
| TB (mg/dl) | | 0.70 (0.50–1.00) | 0.70 (0.50–1.00) |
| AST (IU/l) | | 43 (29–67) | 45 (28–63) |
| ALT (IU/l) | | 56 (35–101) | 48 (30–81) |
| GGT (IU/l) | | 66 (39–106) | 67 (39–116) |
| ALP (IU/l) | | 198 (148–283) | 196 (118–282) |
| Alb (IU/l) | | 4.3 (4.0–4.8) | 4.3 (4.0–4.8) |
| Glycemia (mg/dl) | | 104 (93–126) | 108 (94–129) |
| Creatinine (μmol/l) | | 0.84 (0.71–1.03) | 0.78 (0.64–0.90) |
| Tot Cho (mmol/l) | | 202 (164–246) | 185 (160–227) |
| LDL (mmol/l) | | 107 (83–154) | 112 (85–154) |
| Tg (mmol/l) | | 159 (114–230) | 143 (111–206) |
| FIB-4 | | 1.26 (0.82–2.11) | 1.50 (0.99–2.25) |
| APRI | | 0.53 (0.34–1.03) | 0.53 (0.35–1.01) |
| LS (kPa) | | 8.4 (5.9–13.4) | 9.5 (6.2–14.8) |

n- number, FAST- FibroScan-AST score, Q1- percentile 25, Q3- percentile 75, BMI- body mass index, T2DM- type 2 diabetes mellitus, NASH- nonalcoholic steatohepatitis, CRN- clinical research network, MASH- metabolic disfunction-associated steatohepatitis, Hb- hemoglobin, INR- international normalized ratio, TB- total bilirubin, AST- aspartate-aminotransferase, ALT- alanine-aminotransferase, GGT- gamma glutamyl transpeptidase, ALP- alkaline phosphatase, Alb- albumin, Tot Cho- total cholesterol, LDL- low-density lipoprotein cholesterol, Tg- triglycerides, FIB-4- Fibrosis 4 index, APRI- AST to Platelet Ratio Index, LS- liver stiffness, kPa- kilopascals.

The same trend was respected by Agile 3+ with values for F0 (0.073 ± 0.017), F1 (0.182 ± 0.024), F2 (0.336 ± 0.032), F3 (0.686 ± 0.042) and F4 (0.939 ± 0.016), respectively (Fig 2B).

Congruently, Agile 4 respected the same distribution, with values for F0 (0.006 ± 0.002), F1 (0.024 ± 0.005), F2 (0.100 ± 0.020), F3 (0.292 ± 0.039) and F4 (0.736 ± 0.041), respectively (Fig 2C).

For advanced fibrosis (≥F3), the predictive performance of Agile 3+ was indicated by an AUROC of 0.909 [0.866–0.942] and of Agile 4 by an AUROC of 0.911 [0.869–0.944] (Fig 3). The two scores performances tended to be lower than the AUROC for LSM alone 0.933

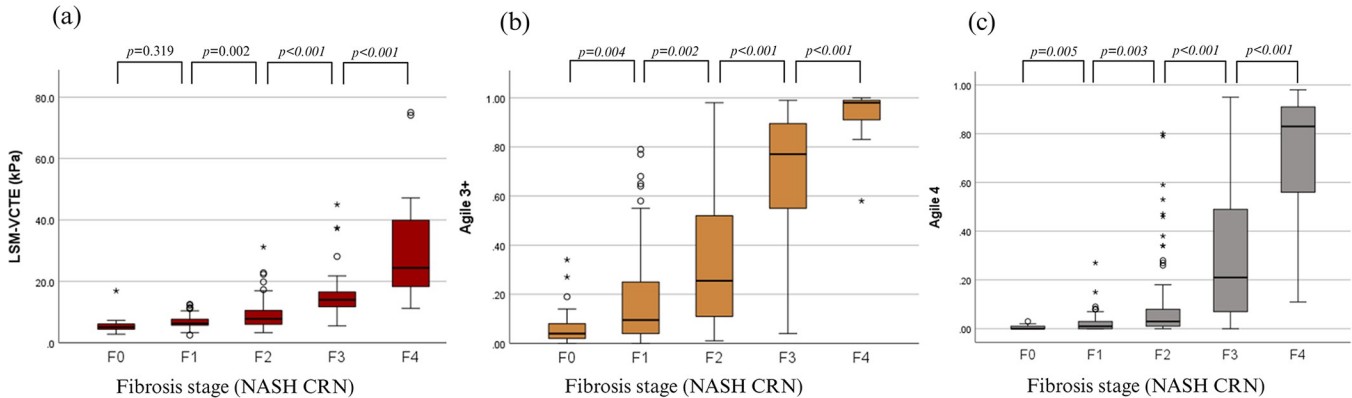

**Fig 2. LSM-VCTE, Agile 3+ and Agile 4 scores among 246 patients with biopsy-proven NAFLD.** Median values (lines inside boxes) for (a) LSM-VCTE, (b) Agile 3+, (c) Agile 4 are shown in the box graph, together with the 25th–75th percentiles, respectively. LSM–liver stiffness measurement, VCTE–vibration controlled transient elastography, kPa–kilopascals, NASH–nonalcoholic steatohepatitis, CRN–Clinical Research network, F0 –absence of fibrosis, F1 –mild fibrosis, F2 –significant fibrosis, F3 –advanced fibrosis, F4 –cirrhosis.

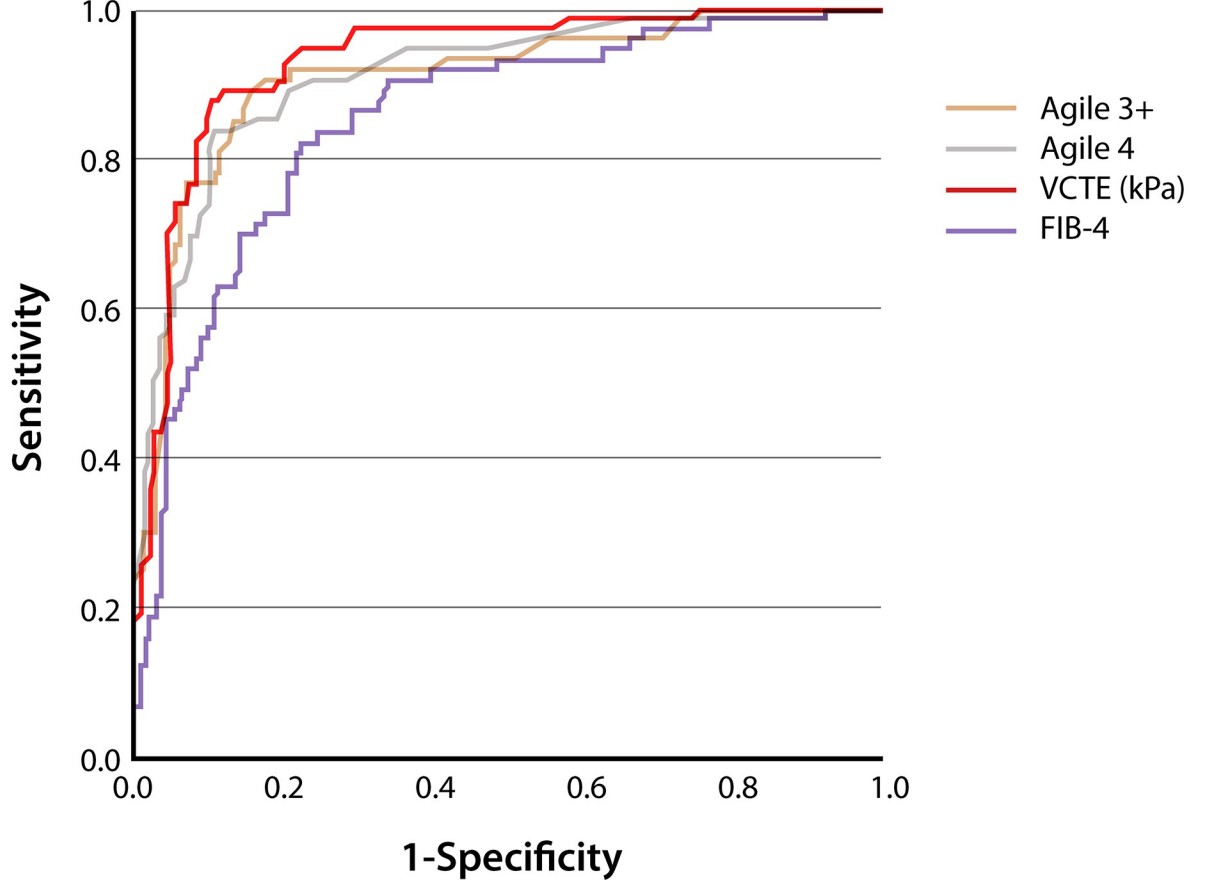

**Fig 3. Diagnostic performance of Agile 3+, Agile 4, LSM-VCTE and FIB-4 in identifying advanced fibrosis (≥F3).** ROC—receiver operating characteristic curve, VCTE–vibration-controlled transient elastography, FIB-4 –Fibrosis 4 Index, kPa–kilopascals.

**Table 2. Diagnostic accuracy of Agile 3+, Agile 4, LSM -VCTE and FIB-4 in identifying advanced fibrosis (≥F3) among 246 patients with biopsy-proven NAFLD.**

| NIT | AUC [95% CI] | Aim | Cut-off | Se (%) | Sp (%) | PPV (%) | NPV (%) | Acc (%) |
|---|---|---|---|---|---|---|---|---|
| **Agile 3+** | 0.909 [0.866–0.942] | Se ≥90% | 0.480 | 90.41 | 82.66 | 68.75 | 95.33 | 84.96 |
| | | Sp ≥90% | 0.680 | 76.71 | 92.49 | 81.17 | 90.40 | 87.81 |
| | | Se ≥85% | 0.530 | 86.30 | 85.55 | 71.59 | 93.67 | 85.77 |
| | | | 0.451* | 90.41 | 79.77 | 65.35 | 95.17 | 82.93 |
| | | | 0.679** | 76.71 | 91.91 | 80.00 | 90.34 | 87.40 |
| **Agile 4** | 0.911 [0.869–0.944] | Youden | 0.090 | 83.56 | 89.60 | 77.22 | 92.81 | 87.81 |
| **VCTE** | 0.933 [0.894–0.961] | Youden | 11.1 kPa | 89.04 | 88.44 | 76.47 | 95.03 | 88.62 |
| **FIB-4** | 0.854 [0.803–0.895] | Youden | 1.53 | 82.19 | 78.03 | 61.22 | 91.21 | 79.26 |

NIT- non-invasive rest, VCTE–vibration controlled transient elastography, FIB-4 –Fibrosis 4 Index, Se–sensitivity, Sp–specificity, PPV–positive predictive value, NPV–negative predictive value, Acc- accuracy, kPa–kilopascals

*Original cut-off value to rule-out advanced fibrosis [10]

** Original cut-off value to rule-in advanced fibrosis [10].

[0.894–0.961], although not statistically significant (DeLong test $p = 0.209$ and $p = 0.245$, respectively).

Using the cut-off of 0.451, Agile 3+ presented a Se = 90.41%, Sp = 79.77%, PPV = 65.35%, NPV = 95.17% and Acc = 82.93% for ruling out advanced fibrosis. By applying the cut-off of 0.679 to rule in advanced fibrosis, Agile 3+ exhibited a Se = 76.71%, Sp = 91.91%, PPV = 80.00%, NPV = 90.34% and Acc = 87.40%, respectively. The performance of Agile 3+ in predicting advanced fibrosis (≥F3) using our best selected cut-offs for Se ≥85%, ≥90% and Sp ≥90% is presented in Table 2.

For cirrhosis (F4), the diagnostic performance of Agile 3+ was indicated by an AUROC of 0.958 [0.925–0.980], and of Agile 4 by an AUROC of 0.968 [0.937–0.986] (Fig 4). The two scores performances tended to be higher than LSM-VCTE alone (0.956 [0.922–0.978]), although not statistically significant (DeLong test $p = 0.782$, and $p = 0.312$, respectively).

Using the cut-off of 0.251, Agile 4 presented a Se = 96.55%, Sp = 84.79%, PPV = 45.90%, NPV = 99.46% and Acc = 86.18% in ruling-out cirrhosis. By applying the cut-off of 0.565 to rule-in cirrhosis, Agile 4 exhibited a Se = 72.41%, Sp = 94.47%, PPV = 63.64%, NPV = 96.24% and Acc = 91.87%, respectively. The performance of Agile 4 in predicting cirrhosis (F4) using our best selected cut-offs for Se ≥90%, Se ≥85%, respectively Sp ≥90%, Sp ≥95% is presented in Table 3.

## Comparison with other fibrosis prediction scores

A detailed comparison using DeLong protocol among different fibrosis prediction scores is presented in Table 4.

In terms of identifying advanced fibrosis (≥F3), the AUROCs for LSM-VCTE alone and for FIB-4 index were 0.933 [0.894–0.961] and 0.854 [0.803–0.895], respectively (Table 2). Agile 3+ had significantly better diagnostic performance compared to FIB-4 ($p = 0.012$), but not compared to LSM-VCTE ($p = 0.209$).

In terms of identifying cirrhosis (F4), the AUROCs for LSM-VCTE alone and for FIB-4 index were 0.956 [0.922–0.978] and 0.921 [0.880–0.951], respectively (Table 3). Agile 4 had a significantly better diagnostic performance compared to FIB-4 ($p = 0.001$), and a slightly better diagnostic performance compared to LSM-VCTE alone, but not statistically significant ($p = 0.312$).

## ROC curves for Agile 3+, Agile 4, VCTE and FIB-4 in identifying cirrhosis

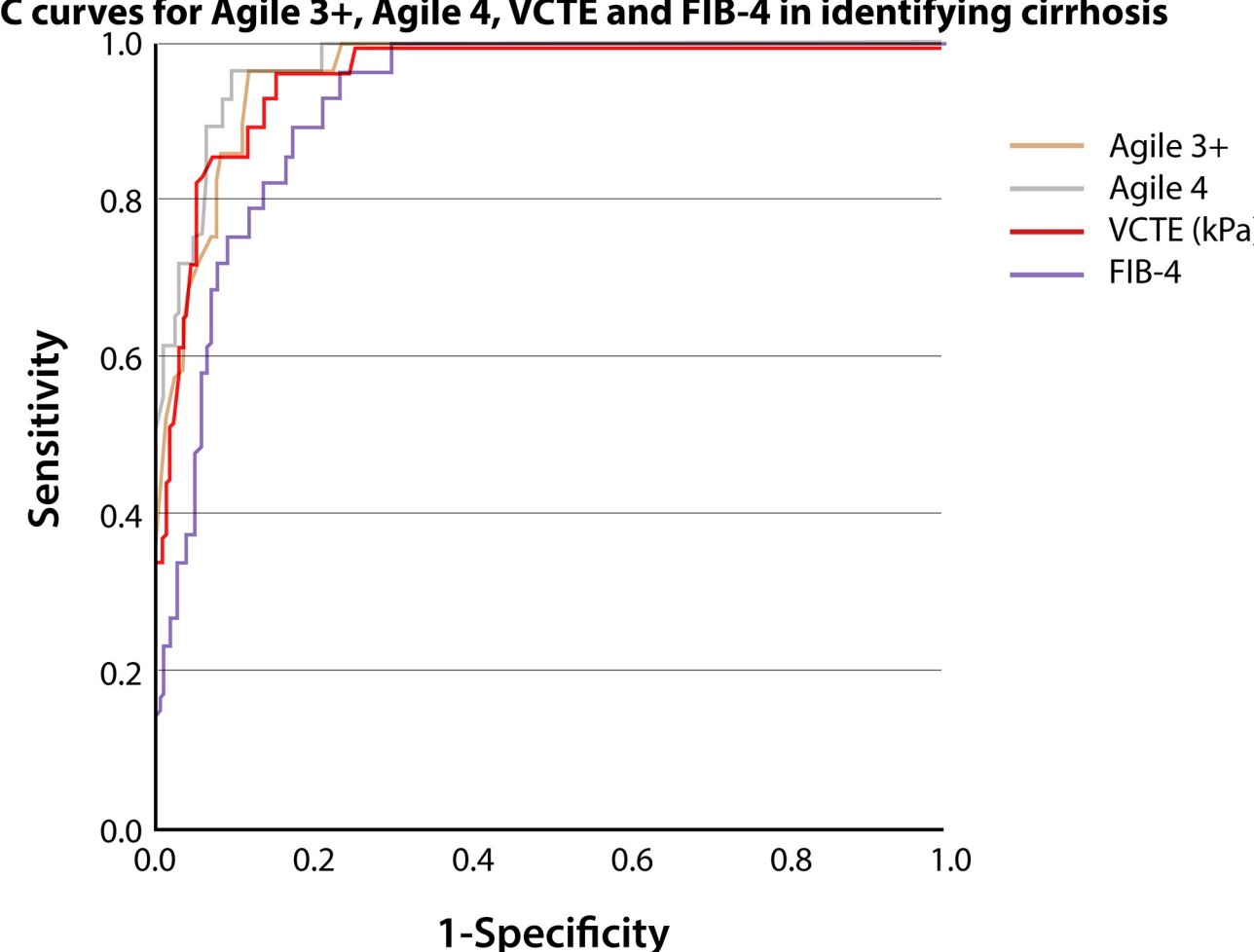

**Fig 4. Diagnostic performance of Agile 3+, Agile 4, LSM-VCTE and FIB-4 in identifying cirrhosis (F4) among 246 patients with biopsy-proven NAFLD.**
ROC—receiver operating characteristic curve, VCTE–vibration-controlled transient elastography, FIB-4 –Fibrosis 4 Index, kPa–kilopascals.

**Table 3. Diagnostic accuracy of Agile 3+, Agile 4, LSM-VCTE and FIB-4 in identifying cirrhosis (F4) among 246 patients with biopsy-proven NAFLD.**

| NIT | AUC [95% CI] | Aim | Cut-off | Se (%) | Sp (%) | PPV (%) | NPV (%) | Acc (%) |
|---|---|---|---|---|---|---|---|---|
| **Agile 3+** | 0.958 [0.925–0.980] | Youden | 0.820 | 96.55 | 87.56 | 50.91 | 99.48 | 88.62 |
| **Agile 4** | 0.968 [0.937–0.986] | Se ≥90% | 0.380 | 93.10 | 90.78 | 57.44 | 98.99 | 91.05 |
| | | Sp ≥90% | 0.520 | 75.86 | 94.01 | 62.86 | 96.68 | 91.87 |
| | | Se ≥85% | 0.470 | 89.66 | 93.09 | 63.42 | 98.54 | 92.69 |
| | | Sp ≥95% | 0.600 | 72.41 | 96.31 | 72.39 | 96.31 | 93.49 |
| | | | 0.251* | 96.55 | 84.79 | 45.90 | 99.46 | 86.18 |
| | | | 0.565** | 72.41 | 94.47 | 63.64 | 96.24 | 91.87 |
| **VCTE** | 0.956 [0.922–0.978] | Youden | 13 kPa | 96.55 | 83.87 | 44.44 | 99.45 | 85.36 |
| **FIB-4** | 0.921 [0.880–0.951] | Youden | 1.79 | 96.55 | 76.04 | 35.00 | 99.40 | 78.46 |

NIT- non-invasive rest, VCTE–vibration controlled transient elastography, FIB-4 –Fibrosis 4 Index, Se–sensitivity, Sp–specificity, PPV–positive predictive value, NPV–negative predictive value, Acc–Accuracy, kPa–kilopascals

*Original cut-off value to rule-out cirrhosis [10]

** Original cut-off value to rule-in cirrhosis [10].

**Table 4. Comparison between Agile 3+, Agile 4, and standard NITs in staging liver fibrosis using DeLong protocol.**

| Fibrosis stage | NIT | p values | | | |
|---|---|---|---|---|---|
| | | Agile 3+ | Agile 4 | VCTE | FIB-4 |
| ≥ F3 | Agile 3+ | N/A | 0.783 | 0.209 | 0.012 |
| | Agile 4 | 0.783 | N/A | 0.245 | 0.003 |
| | VCTE | 0.209 | 0.245 | N/A | 0.006 |
| | FIB-4 | 0.012 | 0.003 | 0.006 | N/A |
| F4 | Agile 3+ | N/A | 0.099 | 0.782 | 0.012 |
| | Agile 4 | 0.099 | N/A | 0.312 | 0.001 |
| | VCTE | 0.782 | 0.312 | N/A | 0.086 |
| | FIB-4 | 0.012 | 0.001 | 0.086 | N/A |

F3 –advanced fibrosis, F4 –cirrhosis, NIT–non-invasive test, VCTE–vibration-controlled transient elastography, FIB-4 –Fibrosis 4 Index, N/A–not applicable.

## Proportion of patients with indeterminate results when applying Agile 3+, Agile 4 and LSM—VCTE

We next looked at the proportion of patients that remained unclassified (in the so-called "grey zone"). In our cohort of patients, by using Agile 3+ standard cut-offs for ≥F3, 0.451 and 0.679 [10], and LSM-VCTE standard cut-offs for ≥F3, 8 kPa and 12 kPa, [11], the proportion of patients that were left unclassified were 12.6% and 21.9%, respectively (McNemar's exact test $p$ = 0.003). By using Agile 4 standard-cutoffs for F4, namely 0.251 and 0.565 [10], and the LSM-VCTE cut-offs of 8 kPa and 20 kPa, then 10 kPa and 20 kPa [19], the proportion of patients that were left unclassified were 11.4%, 39.4% and 26.8%, respectively (McNemar's exact test $p$<0.0001 for both scenarios), as depicted in Table 5.

### Diagnostic performance of FAST score in identifying fibrotic NASH

The AUROCs for FAST Score, LSM-VCTE alone, FIB-4 index and APRI score for identifying fibrotic NASH (NASH + NAS ≥4 + F ≥2) were 0.679 [0.594–0.757], 0.591 [0.503–0.674], 0.519 [0.432–0.606], and 0.578 [0.490–0.662], respectively (Fig 5). In the subgroup analysis for FAST Score, the AUROCs were 0.70 [0.59–0.80] for ALT ≥35 UI/L and 0.60 [0.42–0.77] for ALT <35 UI/L, respectively.

The AUROC for FAST score was significantly higher than the AUROCs for LSM-VCTE alone (p = 0.02), FIB-4 (p = 0.001) and APRI (p = 0.002) scores.

The Se, Sp, PPV, NPV, and Acc for FAST score using the cut-off of 0.35 to rule-out fibrotic NASH and the cut-off of 0.67 to rule in the condition [12], along with our best selected cut-off values for Se ≥90% and Sp ≥90% are depicted in Table 6.

**Table 5. Distribution of patients with biopsy-proven NAFLD according to the individual risk.**

| | NIT | Rule-out cutoff | Patients below the low cut-off value n (%) | Patients remained in grey zone n (%) | Rule-in cutoff | Patients above the high cut-off value n (%) |
|---|---|---|---|---|---|---|
| AF (≥F3) | Agile 3+ | 0.451 | 145 (58.9) | 31 (12.6) | 0.679 | 70 (28.5) |
| | LSM-VCTE | 8 kPa | 120 (48.8) | 54 (21.9) | 12 kPa | 72 (29.3) |
| Cirrhosis (F4) | Agile 4 | 0.251 | 185 (75.2) | 28 (11.4) | 0.565 | 33 (13.4) |
| | LSM-VCTE | 8 kPa | 120 (48.8) | 97 (39.4) | 20 kPa | 29 (11.8) |
| | LSM-VCTE | 10 kPa | 151 (61.4) | 66 (26.8) | 20 kPa | 29 (11.8) |

NIT–non-invasive test, AF–advanced fibrosis, F3 –advanced fibrosis. LSM–liver stiffness measurement, VCTE–vibration-controlled transient elastography, n–number, %—percentage.

## ROC curves for FAST score, VCTE, FIB-4 and APRI scores in identifying fibrotic NASH

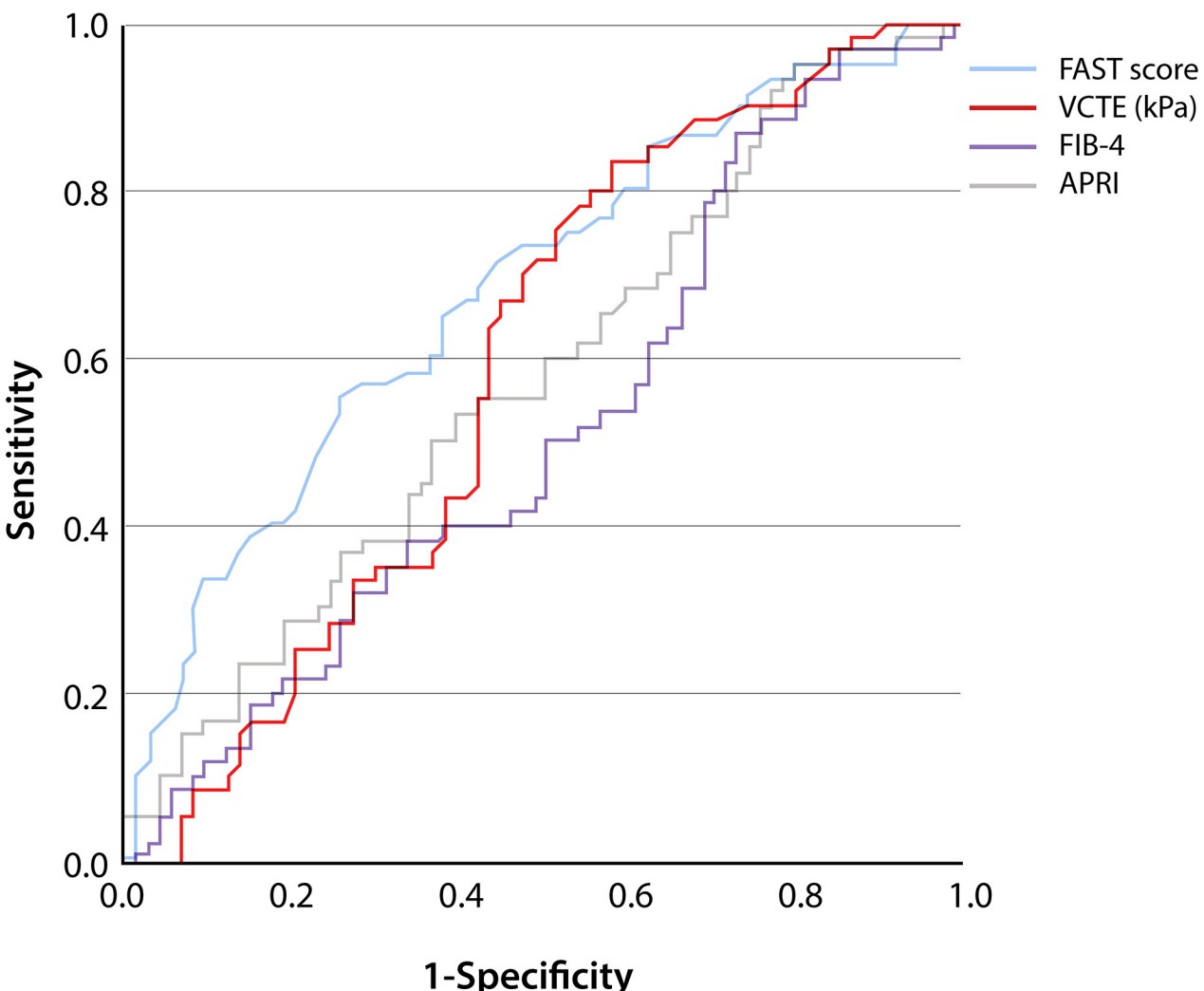

**Fig 5. Diagnostic performance of FAST score and other NITs in predicting fibrotic NASH.** ROC—receiver operating characteristic curve, FAST-FibroScan-AST score, VCTE–vibration-controlled transient elastography, FIB-4 –Fibrosis 4 Index, APRI—aspartate aminotransferase to platelet ratio index.

**Table 6. Diagnostic accuracy of FAST score in predicting fibrotic NASH (NASH + NAS≥4 + F≥2).**

| NIT | AUC (95% CI) | Aim | Cut-off | Se (%) | Sp (%) | NPV (%) | PPV (%) | Acc (%) |
|---|---|---|---|---|---|---|---|---|
| **FAST score** | 0.679 (0.594, 0.757) | Se ≥90% | 0.17 | 91.80 | 25.33 | 79.16 | 50.00 | 55.14 |
| | | Sp ≥90% | 0.75 | 31.15 | 90.67 | 61.82 | 73.09 | 63.97 |
| | | | 0.35* | 21.31 | 58.67 | 47.83 | 41.94 | 41.91 |
| | | | 0.67** | 49.18 | 76.00 | 64.77 | 62.50 | 63.97 |

NIT–noninvasive test, AUC-area under the ROC curve, CI- confidence interval, %- percentage, Se- sensitivity, Sp-specificity, NPV- negative predictive value, PPV-positive predictive value, Acc-accuracy

*original cut-off value to rule-out fibrotic NASH [12]

**original cut-off value to rule-in fibrotic NASH [12].

In our cohort of patients, when applying the FAST score with its standard cut-off values [12], 44 (32.4%) patients remained below the inferior cut-off, 44 (32.4%) in the grey zone and 48 (35.2%) patients were above the superior cut-off.

## Discussion

The purpose of this study was to validate three new non-invasive scoring systems (Agile 3+, Agile 4, and FAST score) in a cohort of 246 patients with biopsy-confirmed NAFLD meeting MASLD criteria. We sought to assess their effectiveness in discriminating advanced fibrosis, cirrhosis, and fibrotic NASH and successfully confirmed their utility. During the validation process, we evaluated the performance of previously published cut-offs, and provided our best selected cut-off values, aiming for sensitivities of ≥85% and ≥90% to rule-out the conditions, and specificities of ≥90% and ≥95% to rule-in the conditions [10].

Agile 3+ and Agile 4 scores were specifically developed for individuals with NAFLD in 2023 [10]. These scoring systems serve three primary objectives: the identification of advanced fibrosis and cirrhosis; the optimization of the positive predictive value to confirm these conditions; and the reduction of cases with indeterminate results, often referred to as the "grey zone" [10].

### Agile 3+

Upon assessing its diagnostic performance, the AUROC for Agile 3+ in discriminating ≥F3 was excellent, but slightly inferior to those for LSM-VCTE alone (0.909 vs. 0.933), even though not statistically significant ($p$ = 0.209). Nonetheless, when comparing the patients with indeterminate results that resulted after applying dual cut-off approaches for Agile 3+ and LSM-VCTE, Agile 3+ significantly reduced the number of patients that remained in the grey zone ($p$ = 0.003), while maintaining very good accuracy (Table 5).

### Agile 4

Upon assessing its diagnostic performance, the AUROC for Agile 4 in discriminating F4 exhibited excellent performance, that was slightly superior to LSM-VCTE alone (0.968 vs. 0.956), although not statistically significant. When applying the dual cut-off approach for Agile 4, only 11.4% of patients remained in the grey zone, while maintaining excellent accuracy. When comparing the patients with indeterminate results that resulted after applying dual cut-off approaches for both Agile 4 and LSM-VCTE, Agile 4 significantly reduced the number of patients that remained in the grey zone (p<0.0001).

Both scores significantly outperformed FIB-4 in discriminating ≥F3 and F4 (Table 4).

These data suggest that Agile 3+ and Agile 4 scores are well optimized to discriminate NAFLD patients with advanced fibrosis and cirrhosis, and our results are consistent with findings from previous reported studies [10, 20–22]. The seminal study by Sanyal et. all. [10], that developed the Agile scores, reported significantly greater AUROCs when compared to LSM-VCTE alone (0.86–0.90 for Agile 3+ and 0.83–0.85 for LSM-VCTE in depicting ≥F3, and 0.89–0.93 for Agile 4 and 0.85–0.88 for LSM-VCTE in discriminating F4). In our cohort of patients, even though we did not obtain significantly greater AUROCs, the Agile 3+ score significantly reduced the number of patients with indeterminant results and Agile 4 exhibited an excellent accuracy of 92%. In our population, by using the superior cut-off of 0.600 for Agile 4, NAFLD-cirrhosis could be ruled-in with an accuracy of 93.5%.

Nevertheless, a possible explanation for the lack of superiority of Agile scores in our cohort of patients in terms of AUROCs (compared to the seminal study) could be attributed to the slightly diverse prevalence of F3 and F4 and disparities in clinical and laboratory data required

for score computation. However, our cohort consisted in consecutive patients with clinical suspicion of having NASH, and therefore the prevalence of F3 and F4 stages is more likely to reflect the distribution in the general population. More than that, our cohort consisted in patients with Caucasian descent, which have a different risk of developing severe fibrosis than Latin-Americans or Hispanics [2]. With these in mind, our results are of highest importance for Central and Eastern Europe, where Caucasian population is prevalent.

In our cohort, the prevalence of advanced fibrosis (≥F3) and cirrhosis (F4) were 29.7% and 11.8% respectively. In the study by Sanyal et al. [10], the prevalence of both conditions was slightly higher, with 54% and 23% of patients presenting ≥F3 and F4, respectively in both training and validation sets, and a similar prevalence in the external validation cohort of 37% and 13%, respectively. The mean age in the Sanyal et al. cohort was 55 ± 16 years for both training and validation cohorts, and a significant proportion of patients presented diabetes, 50.4% in the training and 51% in the validation cohort respectively, with similar proportion for the external validation cohorts. In our study, the median age was 52 years (41–61), and a lower proportion of patients presented diabetes, namely 30.5%.

For Agile 3+, our thresholds—0.480 for ruling out (Se ≥90%) and 0.680 for ruling in (Sp ≥90%) advanced fibrosis—closely mirrored the standard thresholds of 0.451 and 0.679, respectively. Similarly, for Agile 4, our thresholds—0.380 for ruling out (Se ≥90%) and 0.520 for ruling in (Sp ≥90%) cirrhosis—were in line with the literature's proposed thresholds of 0.251 and 0.565, respectively. Given the excellent diagnostic performance demonstrated by the standard cut-offs in our cohort, they can be reliably applied within the Caucasian population.

One notably significant element emphasized in our article is the outstanding capability of Agile 4 in distinguishing cirrhosis (accuracy of 92%), in a population with a median BMI of 29.0 (IQR, 5.1). It was previously established that LSM-VCTE alone ≥25 kPa is adequate for confirming CSPH in non-obese individuals with NASH, but it falls short in the case of obese patients with NASH [23]. In this regard, composite scores with remarkable accuracy, such as Agile 4, could offer significant improvements in depicting CSPH and improve the management of these patients.

As part of the clinical evaluation, especially for risk stratification, the patients that are left in the "grey zone" should undergo, in our opinion, additional monitoring to determine their real fibrosis status. In this scenario, the causes for false positives should be considered, and another non-invasive test could be applied (ELF™, FibroMeter™, FibroTest®) or the patient could undergo liver biopsy in case of discordant NITs [3, 11].

Another notable accomplishment of using Agile scores in clinical practice lies in their ability to predict liver-related events, as recently reported [21, 24, 25]. Since these scores incorporate factors like diabetes [26], which predisposes to hepatic decompensation, and other variables related to prediction of liver-related events (including hepatocellular carcinoma) [27], we anticipate that this field will remain highly dynamic and lively, with continued validation and exploration of the Agile scores.

## FAST-score

When developed, the FAST score exhibited satisfactory performance in both deviation (C-statistic 0.80, 95% CI 0.76–0.85) and validation (C-statistic range 0.85; 95% CI 0.83–0.87) cohorts and was further validated in some populations [12]. For depicting fibrotic NASH, the FAST score presented a satisfactory performance in a recently published meta-analysis that included 12 studies, with an AUROC of 0.79 [28]. By applying the rule-out (≤0.35) and rule-in (≥0.67) cut-offs, 33% remained in the grey zone [28]. In our cohort of patients, the score presented a moderate performance in discriminating fibrotic NASH with an AUROC of 0.679. The score

outperformed FIB-4 (0.679 vs. 0.519), APRI (0.679 vs. 0.578) and LSM-VCTE alone (0.679 vs. 0.591), nevertheless, LSM-VCTE was designed for fibrosis and steatosis assessment only, and the presence of inflammation can significantly impact the results [3].

Given that Agile 3+, Agile 4, and FAST scores identify populations with varied fibrotic and inflammatory statuses, they hold promise for inclusion in algorithms as a screening tool for fibrosis and fibrotic NASH (MASH) [29]. We believe that they could serve as pivotal components within a clinical pathway, perhaps in a "secondary step", as part of evaluations conducted within specialized medical centers. However, it is essential to recognize that these scores include laboratory tests such as platelets, AST, and ALT, commonly encompassed in the scores that are usually applied in a "first step" like FIB-4 or NAFLD fibrosis score. In this respect, we believe that the performance of a multistep algorithm including Agile 3+, Agile 4, or FAST scores should be carefully evaluated and validated in future studies.

The limitations of our study are inherent in its retrospective nature. Because this study was a cross-sectional one, we did not explore the association between the scores and the clinical outcome. Nevertheless, the strengths rely in the fact that we enrolled a relatively large number of patients, and to our knowledge, this is the first report on the use of the Agile scores that originates from East Europe, incorporating mainly Caucasian descents. Furthermore, a single expert pathologist assessed all liver biopsy samples to minimize disagreements among observers in pathological staging. From our cohort of biopsy-proven NAFLD patients, out of 256 patients with reliable VCTE measurements, 252 (98.4%) met the criteria for MASLD definition, illustrating the significance of our results in light of the recent change in definition. Another limitation of the study could be the extended duration over which the analysis was conducted (2007–2023) and the fact that during this period we utilized 2 different Fibroscan devices (FibroScan X1115305, respectively FibroScan® Expert 630 starting with 2016). As the number of biopsies for NAFLD patients from our tertiary center was not very large, we aimed to include as many probes as possible in our analysis. Nevertheless, the protocol has been rigorously followed since its introduction until the present.

## Conclusions

In conclusion, this study successfully validated the utility of three non-invasive scoring systems (Agile 3+, Agile 4, and FAST score) in a cohort of patients with biopsy-confirmed NAFLD, meeting the criteria for MASLD, and of Caucasian origin. The Agile 3+ and Agile 4 scores demonstrated their effectiveness in discriminating advanced fibrosis and cirrhosis, while reducing the number of cases with indeterminate results, and outperforming the FIB-4 score. Although the AUROCs did not significantly exceed those of LSM-VCTE alone, the Agile scores optimized accuracy and decreased the number of indeterminate results. Considering its excellent accuracy in discriminating cirrhosis, the use of Agile 4 score could improve the non-invasive assessment of CSPH in patients with obese NASH (MASH). The FAST score exhibited moderate performance in detecting fibrotic NASH (MASH). Our findings suggest that these scoring systems bring a significant contribution to the assessment and management of patients with MASLD and warrant further exploration in clinical practice.

## Supporting information

**S1 File.**
(RAR)

## Acknowledgments

Figs 1 and 2 were created with BioRender.com.

## Author Contributions

**Conceptualization:** Cristian Tefas.

**Data curation:** Madalina-Gabriela Taru, Cristian Tefas, Lidia Neamti, Iulia Minciuna, Vlad Taru, Anca Maniu, Ioana Rusu, Bogdan Procopet, Horia Stefanescu.

**Formal analysis:** Madalina-Gabriela Taru, Cristian Tefas, Dan Corneliu Leucuta.

**Funding acquisition:** Monica Lupsor-Platon.

**Investigation:** Madalina-Gabriela Taru, Bogdan Procopet, Monica Lupsor-Platon, Horia Stefanescu.

**Methodology:** Madalina-Gabriela Taru, Monica Lupsor-Platon.

**Project administration:** Madalina-Gabriela Taru, Monica Lupsor-Platon.

**Software:** Dan Corneliu Leucuta.

**Supervision:** Cristian Tefas, Monica Lupsor-Platon.

**Validation:** Madalina-Gabriela Taru, Monica Lupsor-Platon.

**Visualization:** Madalina-Gabriela Taru, Lidia Neamti, Iulia Minciuna, Vlad Taru, Bobe Petrushev, Lucia Maria Procopciuc, Dan Corneliu Leucuta, Bogdan Procopet, Silvia Ferri, Monica Lupsor-Platon.

**Writing – original draft:** Madalina-Gabriela Taru.

**Writing – review & editing:** Madalina-Gabriela Taru, Cristian Tefas, Lidia Neamti, Ioana Rusu, Bobe Petrushev, Bogdan Procopet, Silvia Ferri, Monica Lupsor-Platon, Horia Stefanescu.

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
