## [Decision Letter · Decision Letter 0]

10 Mar 2024

PONE-D-24-01692FAST and Agile – the MASLD Drift: Validation of Agile 3+, Agile 4 and FAST Scores in 246 Biopsy-proven NAFLD Patients Meeting MASLD Criteria of Prevalent Caucasian OriginPLOS ONE

Dear Dr. Lupsor-Platon,

Thank you for submitting your manuscript to PLOS ONE. After careful consideration, we feel that it has merit but does not fully meet PLOS ONE’s publication criteria as it currently stands. Therefore, we invite you to submit a revised version of the manuscript that addresses the points raised during the review process.

We look forward to receiving your revised manuscript.

Kind regards,

Jee-Fu Huang, M.D., Ph.D.

Academic Editor

PLOS ONE

“This work was partially funded from the grants number PN-III-P4-PCE-2021-1140 from UEFISCDI, the Romanian Executive Agency for Higher Education, Research, Development, and Innovation Funding and 35167/17.12.2021 from the University of Medicine and Pharmacy “Iuliu Hațieganu” Cluj-Napoca. Madalina-Gabriela Taru is financed by The Study Loans and Scholarships Agency, The Ministry of Education, Romania.”

3. In the online submission form, you indicated that [All the data are available upon reasonable request].

Reviewers' comments:

Reviewer's Responses to Questions

**Comments to the Author**

1. Is the manuscript technically sound, and do the data support the conclusions?

Reviewer #1: Yes

Reviewer #2: Yes

2. Has the statistical analysis been performed appropriately and rigorously? 

Reviewer #1: Yes

Reviewer #2: Yes

3. Have the authors made all data underlying the findings in their manuscript fully available?

Reviewer #1: Yes

Reviewer #2: Yes

4. Is the manuscript presented in an intelligible fashion and written in standard English?

Reviewer #1: Yes

Reviewer #2: Yes

5. Review Comments to the Author

Reviewer #1: This retrospective study aimed to assess the effectiveness of three new, elastography-based

scoring systems for advanced fibrosis ≥F3 (Agile 3+), cirrhosis F4 (Agile 4), and

fibrotic NASH: NASH + NAS ≥4 + F≥2 (FAST score), in 246 patients with biopsy-proven

NAFLD meeting MASLD criteria. The authors concluded that Agile 3+ and Agile 4 are reliable, non-invasive tests for identifying advanced fibrosis or cirrhosis in MASLD patients, while FAST score demonstrates moderate performance in identifying fibrotic NASH. When compared to FIB-4 and LSM-VCTE, both Agile 3+ and Agile 4 performed better than FIB-4 and had a similar performance to LSM-VCTE, but with higher diagnostic accuracy, hence reducing the grey zone.

Comments are as follows

This study could be regarded as an external validation study. It enrolled patients mainly from Caucasian descents and was different from the original study.

1. Line 380. From our cohort of biopsy-proven NAFLD patients, out of 256 patients with reliable VCTE measurements, 246 (98.4%) met the criteria for MASLD definition, …..

The number of patients with reliable VCTE should be 250.

Reviewer #2: General comments

MASLD is a prevalent chronic liver condition with substantial clinical implications. This study aimed to assess the effectiveness of three new elastography based, scoring systems for advanced fibrosis ≥F3 (Agile 3+), cirrhosis F4 (Agile 4), and fibrotic NASH: NASH + NAS ≥4 + F≥2 (FAST score), in a cohort of biopsy-proven NAFLD meeting MASLD criteria. They included 246 Caucasian patients from Romania and confirmed the good performance of the Agile 3+ and Agile 4 scores in identifying >=F3 and F4, and the FAST score demonstrates a moderate performance in identifying fibrotic NASH.

Overall, this is a well-conducted study, and the manuscript is well-written. The results are clinically relevant and useful. Several issues are listed.

Major comments:

1. The patients in this study were enrolled from 2007 to 2023, which is a long period. The accuracy of the measurement of Fibroscan was a concern since the machine had been remodeled afterward. Please describe this issue and its limitations in the discussion.

2. Please compare the difference between the cut-off value proposed in this study and the recommended cut-off from the literature. After your analysis, what will be the cut-off value recommended for clinical suggestions?

3. Because the Agile 3+, Agile 4, and Fast scores identify populations with different fibrotic and inflammation statuses, the authors should provide an algorithm for patients to use the Agile and Fast score as a screening workflow for fibrosis and fibrotic NASH.

4. Please propose a recommendation for patients in the grey zone. Do they need additional monitoring to determine their real fibrosis status?

5. Because liver inflammation (ALT elevation) may impact the LSM-VCTE results. The authors are encouraged to do a subgroup analysis for the performance of the FAST score stratified by ALT levels (eg. ALT < 1x ULN vs. ALT >= 1X ULN).

6. PLOS authors have the option to publish the peer review history of their article (what does this mean?). If published, this will include your full peer review and any attached files.

Reviewer #1: No

Reviewer #2: No

---

## [Author Response · Author response to Decision Letter 0]

18 Apr 2024

We would like to thank you for the opportunity to resubmit our manuscript PONE-D-24-01692 “FAST and Agile – the MASLD Drift: Validation of Agile 3+, Agile 4, and FAST Scores in 246 Biopsy-proven NAFLD Patients Meeting MASLD Criteria of Prevalent Caucasian Origin” after major revisions and we hope the revised and improved version is worthy to be considered for publication in PLOD ONE.

 You will find a point-by-point response to the reviewers’ comments and concerns. 

We alos updated the Funding Statement.

 We included: 

• A rebuttal letter that responds to each point raised by the academic editor and reviewer(s). 

• A marked-up copy of our manuscript that highlights changes made to the original version. 

• An unmarked version of the revised paper without tracked changes. 

Yours sincerely,

Prof. Monica Lupsor-Platon, MD, PhD, on behalf of all co-authors

---

## [Decision Letter · Decision Letter 1]

6 May 2024

FAST and Agile – the MASLD Drift: Validation of Agile 3+, Agile 4 and FAST Scores in 246 Biopsy-proven NAFLD Patients Meeting MASLD Criteria of Prevalent Caucasian Origin

PONE-D-24-01692R1

Dear Dr. Lupsor-Platon,

We’re pleased to inform you that your manuscript has been judged scientifically suitable for publication and will be formally accepted for publication once it meets all outstanding technical requirements.

Kind regards,

Jee-Fu Huang, M.D., Ph.D.

Academic Editor

PLOS ONE

Additional Editor Comments (optional):

Reviewers' comments:

Reviewer's Responses to Questions

**Comments to the Author**

1. If the authors have adequately addressed your comments raised in a previous round of review and you feel that this manuscript is now acceptable for publication, you may indicate that here to bypass the “Comments to the Author” section, enter your conflict of interest statement in the “Confidential to Editor” section, and submit your "Accept" recommendation.

Reviewer #1: All comments have been addressed

Reviewer #2: All comments have been addressed

2. Is the manuscript technically sound, and do the data support the conclusions?

Reviewer #1: Yes

Reviewer #2: Yes

3. Has the statistical analysis been performed appropriately and rigorously? 

Reviewer #1: Yes

Reviewer #2: Yes

4. Have the authors made all data underlying the findings in their manuscript fully available?

Reviewer #1: Yes

Reviewer #2: Yes

5. Is the manuscript presented in an intelligible fashion and written in standard English?

Reviewer #1: Yes

Reviewer #2: Yes

6. Review Comments to the Author

Reviewer #1: (No Response)

Reviewer #2: All of my comments are properly answered by the authors. I have no further questions. 

7. PLOS authors have the option to publish the peer review history of their article (what does this mean?). If published, this will include your full peer review and any attached files.

Reviewer #1: No

Reviewer #2: No

---

## [Editor Report · Acceptance letter]

10 May 2024

PONE-D-24-01692R1 

PLOS ONE

Dear Dr. Lupsor-Platon, 

I'm pleased to inform you that your manuscript has been deemed suitable for publication in PLOS ONE. Congratulations! Your manuscript is now being handed over to our production team.

Kind regards, 

on behalf of

Dr. Jee-Fu Huang 

Academic Editor

PLOS ONE